

# Clostridium butyricum relieve the visceral hypersensitivity in mice induced by Citrobacter rodentium infection with chronic stress

Tengfei Wang[1,2], Lixiang Li[2], Shiyang Li[2], Hongyu Zhao[2], Junyan Qu[2], Yanan Xia[2] and Yanqing Li[1,2,3]

[1] Department of Gastroenterology, Qilu Hospital, Cheeloo College of Medicine, Shandong University, Jinan, Shandong, China
[2] Laboratory of Translational Gastroenterology, Qilu Hospital, Cheeloo College of Medicine, Shandong University, Jinan, Shandong Province, China
[3] Robot Engineering Laboratory for Precise Diagnosis and Therapy of GI Tumor, Qilu Hospital, Cheeloo College of Medicine, Shandong University, Jinan, Shandong Province, China

Corresponding author
Yanqing Li, liyanqing@sdu.edu.cn

## ABSTRACT

**Background**. Visceral hypersensitivity is a common symptom in patients with post-infectious irritable bowel syndrome (PI-IBS), and change of the microbiota is a vital etiological factor of it. *Clostridium butyricum* (*C. butyricum*) is one of the probiotics which is reported as the active components in the treatment of IBS, especially IBS with diarrhea. *Citrobacter rodentium* (*C. rodentium*) is an enteropathogenic bacteria which can produce self-limiting colitis in mice, which have been used to produce a PI-IBS-like mice model. Whether *C. butyricum* could influence the visceral hypersensitivity and gut microbiota of PI-IBS is still unknown. Our study aimed to examine whether the intervention of *C. butyricum* or antibiotics could affect the etiology of visceral hypersensitivity.

**Methods**. C57BL/6 male mice were gavaged with the *C. rodentium* to induce a infective colitis. The *C. butyricum* and antibiotic compound were used to intervene the infected mice 3 days later. A 9-day chronic water avoidance stress (WAS) process was implemented to help induce the visceral hypersensitivity. The abdominal withdrawal reflex (AWR) score was assayed to indicate the visceral hypersensitivity of different groups. On the 7th, 14th, and 30th days after infection, mice feces were collected and high-throughput sequencing was carried out to analyze their gut microbiota.

**Results**. Combined, the *C. rodentium* infection plus chronic stress (WAS) could induce the visceral hypersensitivity in mice. Treatment of the *C. butyricum* after *C. rodentium* infection could relieve visceral hypersensitivity of mice, while no difference was observed in the antibiotic treatment group. The gut microbiota diversity of *C. rodentium* infected mice was similar to the uninfected mice, while there were different microbial communities structure between them. The Shannon and Chao indexes significantly decreased in the antibiotic treatment group compared to other groups at 7th, 14th, and 30th days post-infection, while treatment of *C. butyricum* could maintain the indexes within normal range. At day 14 after infection, the structure of microbiota headed towards normality after the *C. butyricum* treatment. After the WAS, the Shannon and Chao indexes of the control group decreased and the structure of microbiota changed.

The *C. butyricum* treatment could prevent these changes of the gut microbiota induced by WAS.

**Conclusion**. *C. butyricum* could relieve the visceral hypersensitivity in mice induced by *C. rodentium* infection plus chronic stress. It could also remodel the microbiota change caused by the infection and chronic stress. It may be a more effective treatment strategy for PI-IBS than antibiotics.

## INTRODUCTION

Irritable bowel syndrome (IBS) is a common functional gastrointestinal disorder which is defined by chronic abdominal pain or discomfort and altered bowel habits, without histopathologic findings (*Longstreth et al., 2006*). It has a obviously high morbidity (15–23%) across the industrial world, causing a reduced quality of life in patients and a considerable socioeconomic burden (*Guglielmetti et al., 2011*; *Saha, 2014*). Acute infection in the gastrointestinal tract has six-fold risk factor to the development of IBS (*Halvorson, Schlett & Riddle, 2006*). There is a sub-type named post-infectious IBS (PI-IBS) characterized by new occurrence and frequent abdominal pain that exhibits visceral hypersensitivity, with altered bowel habits (mostly diarrhea), following the experience of an acute gastroenteritis episode (*Barbara et al., 2019*). PI-IBS represents about 4% to 36% of all IBS cases (*Spiller & Campbell, 2006*). Patients typically complain of the abdominal pain symptoms which persist for years and the increasing of visceral hypersensitivity is commonly found among them (*Marshall et al., 2010*; *Lee, Annamalai & Rao, 2017*).

Although the pathogenesis of PI-IBS is indistinct and complicated, there is plenty of evidences showing that the intestinal barrier function dysbiosis (*Cotton, Beatty & Buret, 2011*), low grade inflammation of intestinal mucosa (*Spiller et al., 2000*; *Barbara et al., 2004*), neuroendocrine crosstalk (*Ng et al., 2018*), and the changes of the gastrointestinal microbiota (*Lupp et al., 2007*; *Krogius-Kurikka, Lyra et al., 2009*; *Malinen et al., 2005*) are related to it. Remarkably, gut microbial dysbiosis or inflammation plays a key role in the pathogenesis of abdominal hypersensitivity of PI-IBS (*Su et al., 2018*). There are increases in the numbers of *Firmicutes* and *Proteobacteria,* while decrease in numbers of *Bacteroidetes* (*Krogius-Kurikka, Lyra et al., 2009*; *Malinen et al., 2005*). However, there is a relative paucity of research assessing the mechanism between visceral hypersensitivity and the gut microbial remodeling of PI-IBS.

The guidelines of therapeutic strategy for PI-IBS are insufficient, there are still several clinical therapies for the gastroenteritis episode, which can be divided into general treatment (antidiarrheal and fluid supplement) and modifications of the gut microbiome (probiotics and antibiotics) (*Chey, Eswaran & Kurlander, 2015*). The former treatment strategy has little influence on abdominal pain and uncomfortable symptoms, and the treatment of various antibiotic drugs can induce dramatic and irretrievable changes in the gut

microbiota (*Rodino-Janeiro et al., 2018*). On the other hand, probiotics have been reported to be the active component used in this field. With colonized into the intestinal tract and remodeling of the gut microbiota in childhood gastroenteritis occasions (*Freedman et al., 2018*), probiotics have gained a great attention in recent years. The *C. butyricum* is a type of Gram-positive anaerobic bacterium in gut microbiota, and it could regulate intestinal immune function and prevent colitis in mice (*Hayashi et al., 2013*). It was reported that *C. butyricum* can improves symptoms, quality of life and stool frequency in IBS-D patients (*Sun et al., 2018*). Other kind of probiotics such as *Lactobacillus* and *Bifidobacteria* preperations also have potential in reducing abdominal symptoms and regulating bowel habits of IBS-D (*Wang et al., 2014*). Although there is phenomenal evidence for the *C. butyricum* treatment on IBS, whether it could influence the visceral hypersensitivity and gut microbiota of PI-IBS is is still unknown.

It is well known that *Citrobacter rodentium*, which resembles the pathogenic factor pathogens enteropathogenic *E. coli* (EPEC) and enterohaemorrhagic *E. coli* (EHEC) in humans (*Mullineaux-Sanders et al., 2019*), can produce a transient and self-limiting colitis in mice (*Ibeakanma et al., 2009*). It is interesting that the infectious mice treated with the chronic water avoidance test will have induced visceral hypersensitivity.

In this study, we prepared the protocol that we could use mice subjected to the *C. rodentium* infection plus chronic water stress as the rodent model of PI-IBS. According to the above-mentioned research results, our idea is that intervention of *C. butyricum* may actually involve in or influence the etiology of visceral hypersensitivity in PI-IBS through microbiotia modification.

## MATERIALS & METHODS

### Mice model

Thirty C57BL/6 male mice (5–6 weeks old, 15–20 g) were purchased from SPF Biotechnology Co., Ltd. (Beijing, China), and twenty-four of them were randomly assigned to four groups: the *C. rodentium* gavaged group (*C. rodentium* group), the *C. rodentium* + *C. butyricum* intervention group (*C. butyricum* group, or probiotic group), the *C. rodentium* + antibiotics group (antibiotic group) and the control group with PBS gavaged group ($n = 6$ per group). The *C. rodentium* infected groups were gavaged with *C. rodentium* ($2 \times 10^9$ colony forming units (CFU), 200 μl of PBS) on day 0. Mice in the probiotic and antibiotic groups were gavaged with *C. rodentium* together on day 0, while *C. butyricum* or antibiotics were joined in 3 days later. Mice in the probiotic group were gavaged *C. butyricum* ($1 \times 10^9$ CFU, 200 μl of PBS) daily for one week. Mice in the antibiotic group were given gentamycin (Sigma, St. Louis, MO, USA) at 20 mg/kg/d and ampicillin (Sigma) at 500 mg/kg/d in drinking water for one week (Fig. 1). The rest of the six mice were given *C. rodentium* and deeply euthanizing with 5-minutes $CO_2$ asphyxiation at 30% Chamber Replacement Rates at day 0 ($n = 2$), day 5 ($n = 2$) and day 21 ($n = 2$) after infection, and their distal colonal tissues were collected for HE Staining. Mice that survived the study were bred for our other experiments. All mice were maintained on a 12-h light/dark cycle, with room temperature of 22 °C–24 °C and relative humidity of 50–60%. Mice were

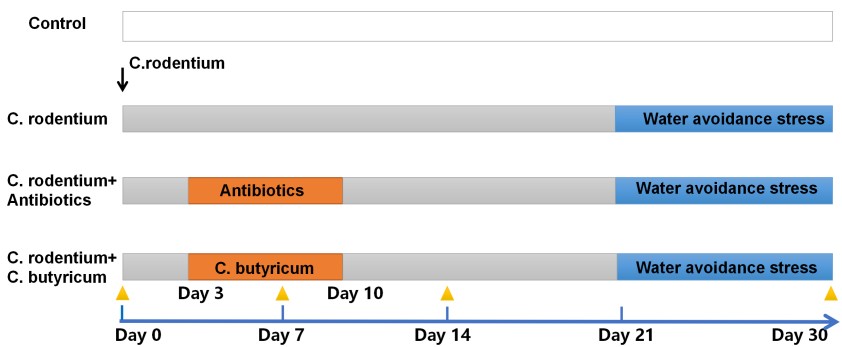

**Figure 1** **Timeline for** *C. rodentium* **infection and** *C. butyricum***/antibiotics intervention for mice.** *C. rodentium* gavage began at 0 days to 5–6 weeks old mice except the control group. The *C. rodentium* + *C. butyricum* intervention group and the *C. rodentium* + antibiotics group were given the probiotics or antibiotics, respectively. And the intervention last for 1 week (orange stripes). Three *C. rodentium* infectious groups were followed by water avoidance stress (WAS) for a period of 9 days (blue stripes). Mice feces samples were taken and stored weekly (yellow triangles) until day 30.

allowed aseptic food and water in a level 2 animal feeding facility at Shandong University Laboratory Animal Center, and we have mixed the bedding for all mouse cages randomly at the beginning of the mice experiment. All procedures in this experiment were in accordance with the ARRIVE guidelines, and were approved by the Shandong University Laboratory Animal Center and the Ethics Committee on Animal Experiment of Shandong University Qilu Hospital (animal experiment proof certificate number: Dull-2020-11).

## Culture of *C. rodentium* and *C. butyricum*

A single *C. rodentium* (ATCC51459) colony was inoculated into Luria broth (LB) medium and grown overnight (at 37 °C and 150 rpm for 12 h). The *C. butyricum* was seperated from *C. butyricum* capsules (ATaiNing, Qingdao Eastsea Pharmaceutical Co., Ltd., China, 420 mg per capsule, $1.5 \times 10^7$ CFU/g, CGMCC0313.1) and cultured anaerobically at 37 °C for 14 h in De Man, Rogosa and Sharpe (MRS) medium.

## Water-avoidance stress (WAS)

The *C. rodentium* infected mice group, the probiotic group, and the antibiotic group were all given a 9-days chronic WAS paradigm (days 21- 30 after infection). For 1 h each day of the chronic stress period, mice were placed on a dry platform (70 mm diameter) in an acrylic bucket of water (280 mm diameter), standing 20 mm above water level and 10 mm below the brim of the bucket. Mice were stressed within a single period of chronic stress (9 days of WAS concluding on post-gavage day 21) as shown in Fig. 1.

## Behavioral testing

Behavioral responses to CRD were assessed in four groups starting 1 days after the water-avoidance stress by measuring the abdominal withdrawal reflex (AWR) using a semiquantitative score. The testing of AWR indicated an express contraction in the abdominal wall musculature. Distention water balloons (6-Fr, two mm external diameter) were placed in the descending colons of mildly sedated mice (4% isoflurane, temporary

inhaled anesthesia) and secured by taping the mouse's tail, and then they were put into a transparent acrylic box. The mice were then housed in small boxes as a recovery room and allowed to wake up and adapt (1 h). Measurement of the AWR consisted of visual observation of the animal response to graded CRD (0.1, 0.15, 0.2, 0.25 and 0.3 ml of water injected into balloons) by double-blinded observers and assignment of the AWR score. Each mouse was tested three times at each graded CRD and averaging for analysis.

## HE staining

The HE staining was conducted in the colonic histology specimens of the six *C. rodentium* infected mice. The paraffin-embedded distal colonic tissues were cut into 4 μm longitudinal sections, after proper specimen processing which involves dehydration, clearing, and paraffin infiltration. After deparaffinization and rehydration, the sections were stained with hematoxylin solution for 2 min followed by 3–4 drips of 1% acid ethanol (1% HCl in 70% ethanol) for 2 s. After the rinse by water, they were stained with eosin solution for 2–3 min and followed by dehydration with graded alcohol and xylene. The slides were observed and photographed using a fluorescence microscope (Nikon TI-FLC-E; Nikon,Tokyo, Japan).

## 16S gene pyrosequencing analysis of mice fecal samples

At the point-in-time of orange triangles in Fig. 1, feces samples were collected at day 0 and in each week after *C. rodentium* infection and they were weighed and homogenized in sterile PBS for microbiota analysis. Fecal samples from the four group mice were collected and DNA was extracted, and the 16S rRNA gene was amplified and sequenced. The 16S rRNA pyrosequencing was processed at Majorbio (Shanghai, China) by using the Illumina Miseq system. The Chao index and Shannon index were calculated to assess the alpha-diversity in each sample. Cluster analysis based on the Euclidean distance was conducted based on the relative abundances of all operational taxnomic units (OTUs). The principal co-ordinates analysis (PCoA) based on the Bray-Curtis distance was done in order to performed to assess the beta-diversity.

## Data analysis

Data are expressed as mean ± SD, and error bars in figures represent SD. The statistical analysis of AWR scores was performed using paired Student's t-tests, and significance was assigned when $*P < 0.05$, $**P < 0.01$, $***P < 0.001$, $\#\#P < 0.01$. The raw data of 16S gene pyrosequencing were clustered into OTUs based on a 97% similarity and analyzed through RDP classifier 2.11. The community composition at each taxonomic level was calculated by Bayes algorithm. The OTUs were then imported into Mothur v 1.38.1 for analyzing the rarefaction curves and alptha diversity. The specific taxa were identified by the Linear discriminant analysis Effect Size (LEfSe) analysis with the value of Kruskal -Wallis sum-rank test set to 0.05. The taxonomy of each 16S rRNA gene sequence was analyzed by RDP Classifier algorithm (http://rdp.cme.msu.edu/) against the Silva (SSU132) 16S rRNA database using confidence threshold of 70%. SPSS was used to analyze the data, a 95% confidence level, $P < 0.05$ was used.

## RESULTS

### *C. rodentium* infection plus repeated water-avoid stress caused the visceral hypersensitivity in mice

All mice exposed to *C. rodentium* alone showed signs of colitis early in the infection (Day 1 to Day 4), such as soft stools, rough body hair, languid movement and temporary body weight loss (Day 2 to Day 5). Two-week post infection, these symptoms gradually faded away, and all mice finally recovered from infectious colitis and brought into next experiments. On the 30th days after *C. rodentium* infection and WAS treatment, the AWR score showed that the infectious group had a lower pain tolerance for colorectal distension compared with the control group ($n = 6$, $P < 0.01$) as shown in Fig. 2A. Also the colonic inflammation pathological changes of the infectious group had recovered to normal within the assessment of the histopathological score (Fig. 2B). Based on these phenomena, we can draw the conclusion that *C. rodentium* infection and AWR stress could produce a PI-IBS-like mice model.

### *C. butyricum* could relieve the visceral hypersensitivity of PI-IBS-like mice

We evaluated whether the visceral hypersensitivity caused by *C. rodentium* infection and WAS could be influenced with probiotics or antibiotics treatment. It was observed that mice treated with the *C. butyricum* presented lower AWR scores in the behavioral testing than *C. rodentium* infected mice group ($n = 6$, $P < 0.05$). However, the gentamycin-ampicillin treatment group did not show this characteristic ($n = 6$, $P > 0.05$) (Fig. 2A).

### The OTUs of gut microbes tremendously decreased after antibiotics treatment,which was not observed in *C. butyricum* treatment mice

The 16S rRNA sequence was performed to investigate the change of the gut microbiota. It was found that the number of OTUs were among 661–673 at the beginning in different groups. The *C. rodentium* infection and *C. butyricum* treatment did not affect the number of OTUs. However, the number of OTUs obviously decreased to 580 at day 7 and further tremendously decrease to 122 at day 14 in antibiotic treatment group (Fig. 3B). After WAS, the number of OTUs in antibiotic group decreased to 111 at day 30 (Fig. 3B). At day 30, there were only 69 OTUs which were shared in the four groups. In contrast, there were 456 OTUs which were shared in the three groups except for the antibiotics group (Fig. 3A). These results demonstrated the antibiotic indeed destroy the abundance of microbiota in PI-IBS-like mice, while *C. butyricum* did not influence it.

### The $\alpha$-diversity of gut microbiota obviously changed after antibiotics treatment, and the *C. butyricum* could maintain them within normal range

The $\alpha$-diversity of gut microbiota was analyzed and the results are shown in Fig. 4. It was found that there was no difference in the Shannon and Chao indexes at day 0 (data not shown). After *C. rodentium* infection, the Shannon and Chao indexes were not affected at day 7, day 14 and day 30. The treatment of *C. butyricum* also did not affect the Shannon and Chao indexes. However, the Chao index of the antibiotics group significantly decreased

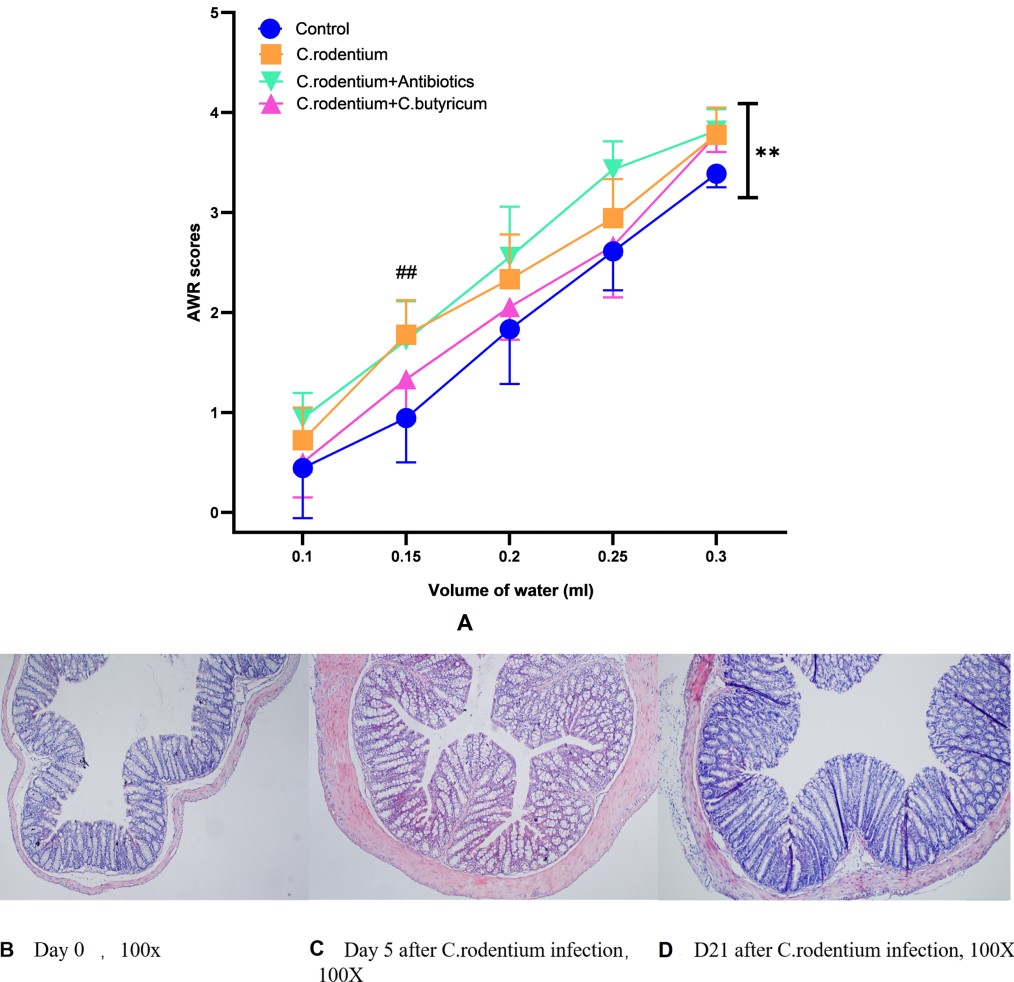

**B** Day 0 , 100x     **C** Day 5 after C.rodentium infection, 100X     **D** D21 after C.rodentium infection, 100X

**Figure 2** **The AWR score and H&E-stained distal colon sections of mice.** (A) Summary data ($n = 6$ mice per group) illustrating the AWR scores which were the responses to graded CRD with water injection in mice subjected to the PBS-only control group (blue circles), *C. rodentium*-only (orange squares), *C. butyricum* co-treatment with *C. rodentium* (pink triangles), and gentamicin-ampicillin co-treatment with *C. rodentium* (green triangles). The AWR scores were higher in the WAS plus *C. rodentium* infectious group compare with control group (** $P < 0.01$), and the co-treatment of *C. butyricum* could reverse this tendency (* $P < 0.05$) while co-treatment of antibiotics had no such effect on it ($P > 0.05$). When the water injected into balloons was 0.15 ml, the AWR scores of *C. rodentium* group was apparently higher than control group (## $P < 0.01$). Data are reported as mean ± SD; paired Student *T*-test. (B–D) H&E-stained distal colon sections from *C. rodentium* infectious mice at 0 day (B), 5 days (C), and 21 days (D) after infection. No inflammation is evident in sections of untreated mice and 21 days after infection mice, while there was inflammatory cell infiltration and hyperplasia in the mucosa of *C. rodentium* infectious mice at day 5.

compared with the control group ($519.45 \pm 52.71$ VS. $436.72 \pm 36.68$, $P < 0.05$) at day 7. Also, both the Shannon and Chao indexes of the antibiotics group were significantly lower than those of the *C. butyricum*-treatment group ($3.45 \pm 0.41$ VS $4.26 \pm 0.41$, $P < 0.01$; $436.72 \pm 36.68$ VS. $524.12 \pm 91.84$, $P < 0.05$). At day 14, the Shannon and Chao indexes of the antibiotics group decreased and were significantly lower than those of the other three

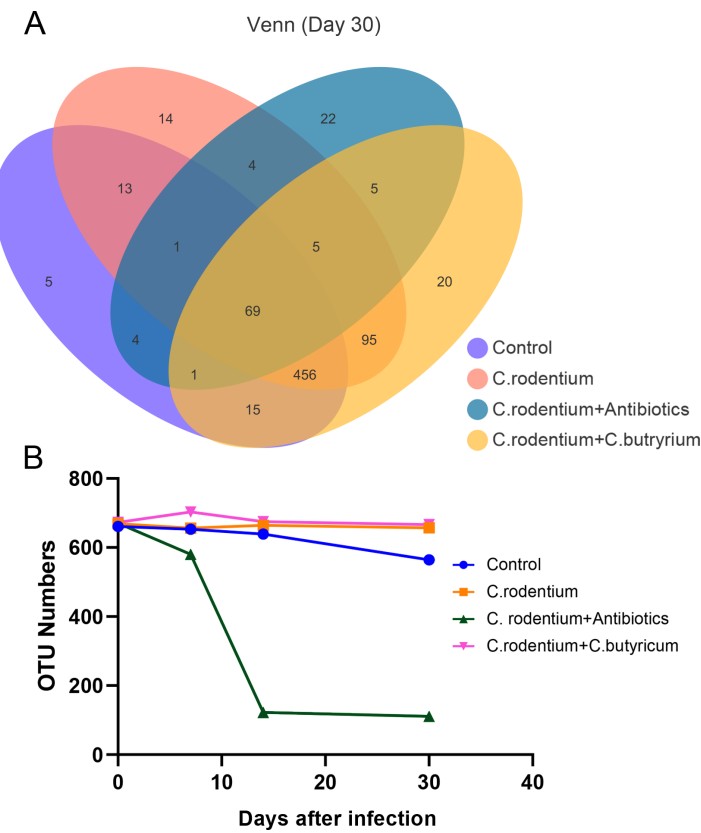

**Figure 3** **Comparison of the OTU of the microbiota in different groups.** (A)The 16S rRNA sequence for the number of OTUs plotted to the Venn diagram of the four experimental groups at day 30 after WAS. There were only 69 OTUs which were shared in the four groups. In contrast, there were 456 OTUs which were shared in the three groups except for the antibiotics group. (B) OTU numbers were counted in the four groups at different times (day 0, day 7, day 14 and day 30). The antibiotic treatment after *C. rodentium* infection demonstrated a low bacterial species diversity compared with control and other infectious groups.

groups ($P < 0.001$). At day 30, the Shannon and Chao indexes of the antibiotics group were still significantly lower than those of the *C. rodentium* infection group and the *C. butyricum*-treatment group ($P < 0.001$).

### The β-diversity of the gut microbiota changed in different groups

The β-diversity of the gut microbiota was analyzed based on PCoA and the results are shown in Fig. 5. The structure of the gut microbiota was nearly the same at day 0 ($R = 0.01$, $P = 0.41$). It was found that *C. rodentium* infection did not affect the structure of the gut microbiota compared with the control group ($R = 0.16$, $P = 0.09$). However, antibiotic treatment significantly affected the structure of the gut microbiota at day 7 ($R = 0.32$, $P = 0.001$) and day 14 ($R = 0.55$, $P = 0.001$). At day 14, the structure of the microbiota was similar between the control and the *C. butyricum* groups. However, there were significant differences between the four groups ($P < 0.05$) (Fig. S1). After the WAS, the difference

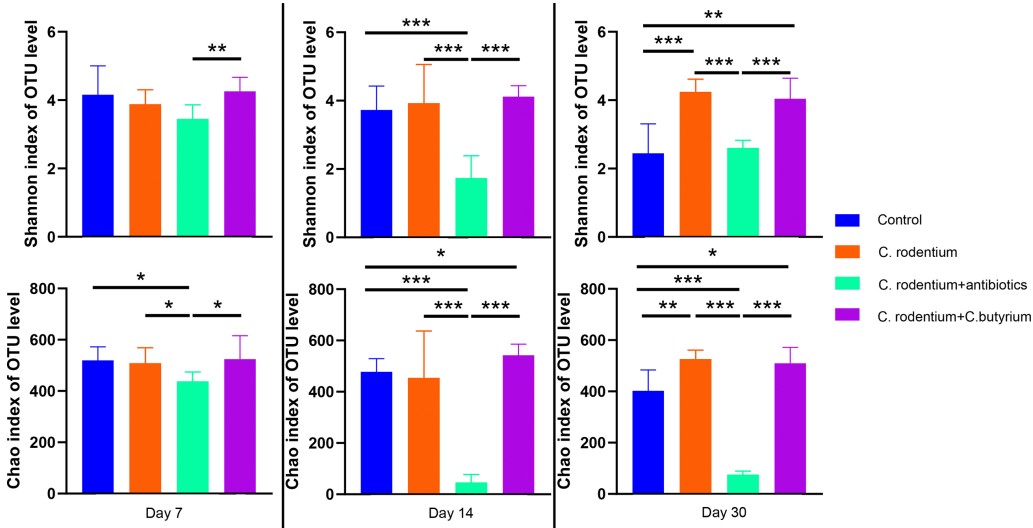

**Figure 4  The α-diversity of gut microbiota in the four groups at different times.** The Chao index of the antibiotic group decreased compared with the control group ($519.45 \pm 52.71$ VS. $436.72 \pm 36.68$, $P < 0.05$) at day 7; both the Shannon and Chao indexes of the antibiotic group were also lower than the *C. butyricum*-treatment group ($3.45 \pm 0.41$VS$4.26 \pm 0.41$, $P < 0.01$; $436.72 \pm 36.68$VS$524.12 \pm 91.84$, $P < 0.05$) at day 14; at day 30, the Shannon and Chao indexes of the antibiotic group were still lower in antibiotic group compared with the other three groups ($P < 0.001$). The *C. butyricum*-treatment group did not affect the Shannon and Chao indexes at day 7, day 14, and day 30.

between the antibiotic treated group and the other three groups was further increased ($R = 0.58$, $P = 0.001$) (Fig. S2).

## The microbiota composition changed in different groups

The main genera of the gut micriobiota (the percentages were above 1%) included more than 38 genera in all groups including *Muribaculaceae, Lactobacillus, Bacteroides, Lachnospiraceae, Ileibacterium, Alistipes,* and *Helicobacter,* ect. (Fig. 6A). The specific taxa that most likely contributed to the differences between the four groups at day 30 were determined by linear discriminant analysis effect size (Fig. 6B). The genera *Lactobacillus, Turicibacter* and *Flavobacteriaceae* were enriched in the control group while the genera *Muribaculaceae, Lachnospiraceae, Helicobacter, Ileibacterium, Alistipes,* and *Ruminococcus* were enriched in the *C. rodentium* infection group. After treatment with antibiotics, the genera *Bacteroides, Blautia, Parasutterella, Parabacteroides,* and *Akkermansia* were enriched. In contrast, the genera *Oscillospirales, Mucispirillum, Dubosiella,* and *Odoribacter* were enriched in the *C. butyricum*-treatment group. The amount of genera *Lactobacillus* and *Odoribacter* in the four groups are shown in Figs. 6C and 6D. It was found that the amount of the genus *Lactobacillus* in the control group was higher than those of other groups ($P < 0.01$). It increased in *C. butyricum*-treatment group without significant difference. Furthermore, the genus *Odoribacter* in the *C. butyricum*-treated group was higher than those of the other three groups ($P < 0.05$).

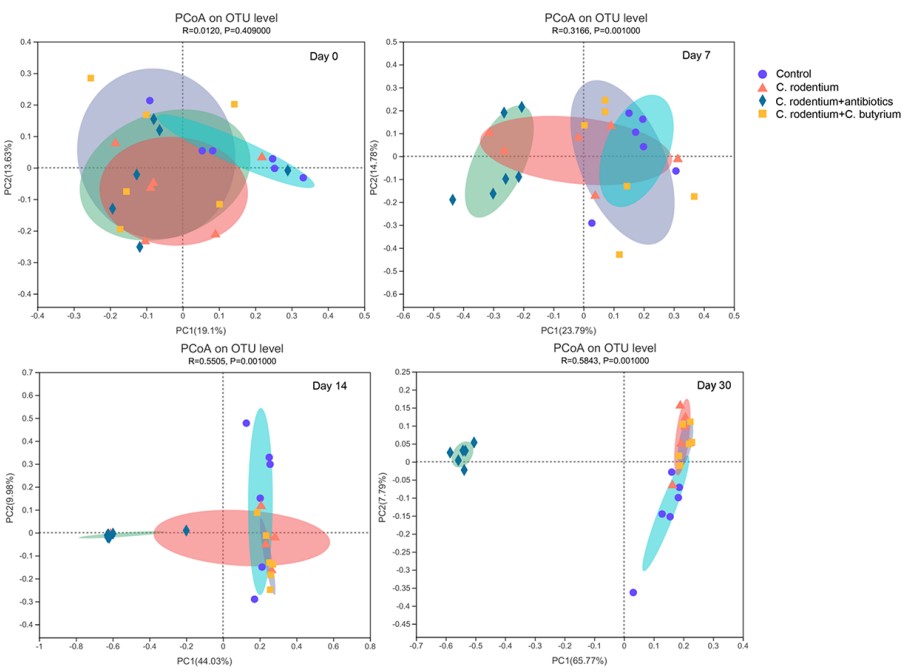

**Figure 5** **The $\beta$-diversity of the gut microbiota in the four groups was analyzed based on PCoA at different times.** The structure of the gut microbiota was nearly the same at day 0 ($R = 0.01$, $P = 0.41$). *C. rodentium* infection did not affect the structure of the gut microbiota compared with the control group ($R = 0.16$, $P = 0.09$). However, antibiotics treatment significantly affected the structure of the gut microbiota at day 7 ($R = 0.32$, $P = 0.001$) and day 14 ($R = 0.55$, $P = 0.001$). After the WAS, the difference between the antibiotic treated group and the other three groups was further increased ($R = 0.58$, $P = 0.001$).

## DISCUSSION

In this study, after treatment with *C. butyricum*, the PI-IBS mouse model presented a lower AWR scores which means mitigatory peritoneal hypersensitivity. In contrast, antibiotics treatment had no effects on hypersensitivity. We also found that *C. butyricum* treatment could remodeling of the gut microbiota. However, antibiotics treatment could induce a drastic change in the intestinal microbiota.

In recent years, it was reported that lots of probiotic supplements have favorable effects on IBS as revealed through in clinical cases and animal studies (*Ford et al., 2014*; *Didari et al., 2015*). As the most commonly used probiotics, *Bifidobacterium* is of great benefit to on abdominal pain and bloating in IBS patients (*Sun et al., 2018*). *Lactobacillus acidophilu* s improve an improvement of barrier function and reduce cytokines secretion, which contribute to resulting in the benefit of relieving visceral hypersensitivity of IBS (*Zeng et al., 2008*). There was also a PI-IBS rodent model study to induce colitis through the *Trichinella*, and the intervention of *Lactobacillus* and yeast showed a decrease of visceral hypersensitivity in mice (*Hong et al., 2019*). The *Trichinella* causes persistent and histoclastie infection in gut tissue (*Luo et al., 2019*), which may not reflect the pathogenic condition of PI-IBS. In this study, the infection of *C. rodentium* is a self-limited colitis in mice which have a histological recovery in 10 days after infection (*Collins et al., 2014*).

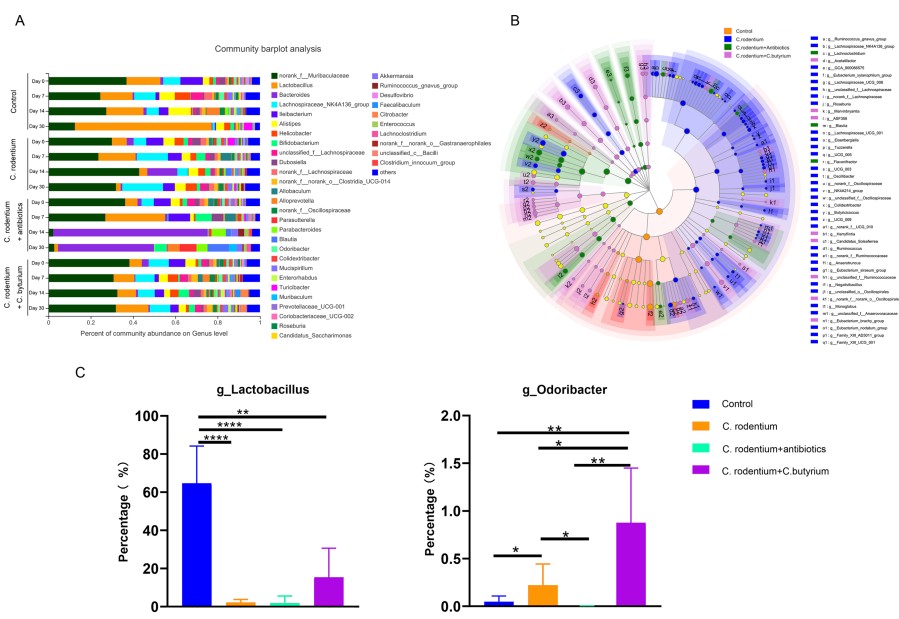

**Figure 6  The microbiota composition changed in different groups.** (A) The main genera of the gut microbiota (the percentages were above 1%) included more than 38 genera in all groups. (B) The specific taxa that most likely contributed to the differences between the four groups at day 30 were determined by linear discriminant analysis effect size. (C) The amount of genera *Lactobacillus* and *Odoribacter* in the four groups were all increased in the *C. butyricum*-treatment group compared with other two *C. rodentium* infectious groups. The genus *Lactobacillus* in the control group was higher than those of other groups. (* $P < 0.05$, ** $P < 0.01$, *** $P < 0.001$, **** $P < 0.0001$).

This rodent model could mimic a similar pathological and immunological state of PI-IBS cases. The *C. butyricum* is a well-known probiotic which was used for therapeutic strategy for IBS patients (*Sun et al., 2018*; *Zhao et al., 2019*). It revealed an important role in a hypersensitivity-relieving effect in the *C. rodentium* infection plus WAS mice model (Fig. 2A). Antibiotics are commonly used in the treatment of infectious diarrhea. In this study, an antibiotic mixture of gentamycin-ampicillin was also used to treat the *C. rodentium* infection. Though the fecal character greatly changed, it demonstrated no significant differences in hypersensitivity of infected mice plus WAS (Fig. 2A). Thus, the *C. butyricum* might be a better choice for infectious diarrhea than antibiotics.

There have been many studies on the pathogenesis of IBS, and one of the most important explains is gut microbiota remodeling. Gastroenteritis episodes induce variation in the gut microbiota, which usually return to normal in healthy people after recovery. However, those developing PI-IBS may have a relative inability to restore their microbiota (*Neal, Barker & Spiller, 2002*). It was reported that the C57BL/6 mice used in our study is one of *C. rodentium*-resistant mouse line which could only get a self-limiting colitis and more mild symptoms of diarrhea and weight loss compare to the sensitive mouse lines (*Vallance et al., 2003*). The amount of *C. rodentium* were only about $10^2$–$10^4$/g feces 3 days after infection in the resistant mice (*Osbelt et al., 2020*). And the $\alpha$-diversity remained stable and non-significant decrease during the infection until day 14 (*Cannon et al., 2020*). However,

the intervention of *C. rodentium* did affect the structure of gut microbiota same as the published reports (*Osbelt et al., 2020*; *Hoffmann et al., 2009*). In this study, the structure of the microbiota was still not restored at day 14 after infection without treatment or treatment with antibiotic (Fig. 5 and Fig. S1). In contrast, the structure of the microbiota of the mice in the *C. butyricum* treatment group was similar to those of the control group (Fig. S1), which indicated its effect on maintaining the balance of gut microbiota. Also, chronic persistent stress, such as WAS, plays an important role in remodeling the gut microbiota of mice (*Han et al., 2020*).

In this study, we found that WAS treatment remodeled the microbiota and the Shannon and Chao indexes even decreased in control group (Fig. 3). However, the Shannon and Chao indexes did not change after WAS treatment in the *C. butyricum* treatment group. It was interesting that only the structure of the microbiota of the *C. butyricum* treatment group did not change after WAS (Fig. S2). This suggested that *C. butyricum* could protect the microbiota from chronic persistent stress. Furthermore, the genera *Odoribacter* and *Lactobacillus* increased more in the *C. butyricum* treatment group at day 30 than in the *C. rodentium* infections group and in the antibiotic group. It was reported that higher baseline proportions of *Odoribacter* were related to beneficial inflammatory-marker changes (*Hod et al., 2018*). In addition, probiotics of the genus *Lactobacillus* are also have potentially reduced abdominal symptoms of IBS (*Bonfrate et al., 2020*; *Preston et al., 2018*). These may be the reason why *C. butyricum* alleviates visceral hypersensitivity.

On the other hand, the antibiotics can affect the gut microbiota of PI-IBS (*Becattini, Taur & Pamer, 2016*). In this study, we found that the Shannon and Chao indexes as well as the structure of the microbiota significantly changed after the treatment of antibiotics (Figs. 4 and 5) and the tendencies could last for 30 days. It was also found that the genus *Bacteroides* increased at day 14 and 30 (Fig. 6A) in antibiotic group. The increase of the *Bacteroidetes* phylum composition may increase the susceptibility to infection and it is also abundant in PI-IBS patients (*Downs et al., 2017*). Therefore, probiotics are more suitable than antibiotics in remodeling the microbiota of patients after infection.

There may be some possible limitations in this study. First, whether or not probiotics include other strains of *Clostridium* have the same effect on the visceral hypersensitivity induced by the intestinal infection was not confirmed. Second, the PI-IBS-like rodent model can not identically imitate the pathological conditions of PI-IBS patients, it need to continually improve. Thirdly, we observed the *C. butyricum* could relive the visceral hypersensitivity and the gut microbiota was closely related to it, the specific molecular or immune mechanism was not explored and need further research.

## CONCLUSIONS

In conclusion, it was found that the *C. butyricum* could relieve the visceral hypersensitivity in mice induced by *C. rodentium* infection plus chronic stress. It was also superior to

antibiotics in remodeling the microbiota of the visceral hypersensitivity model mice. Thus, *C. butyricum* may be a candidate for the treatment of PI-IBS.

### Funding

This work was supported by the National Natural Science Foundation of China (81873550), Key Research and Development Program of Shandong Province (2019GHZ022). This study is also funded by the Taishan Scholars Program of Shandong Province and National Clinical Research Center for Digestive Diseases supporting technology project (2015BAI13B07). The funders had no role in study design, data collection and analysis, decision to publish, or preparation of the manuscript.

### Grant Disclosures

The following grant information was disclosed by the authors:
The National Natural Science Foundation of China: 81873550.
Key Research and Development Program of Shandong Province: 2019GHZ022.
The Taishan Scholars Program of Shandong Province and National Clinical Research Center for Digestive Diseases: 2015BAI13B07.

### Competing Interests

The authors declare there are no competing interests.

### Author Contributions

- Tengfei Wang conceived and designed the experiments, performed the experiments, analyzed the data, prepared figures and/or tables, authored or reviewed drafts of the paper, and approved the final draft.
- Lixiang Li conceived and designed the experiments, analyzed the data, prepared figures and/or tables, authored or reviewed drafts of the paper, and approved the final draft.
- Shiyang Li analyzed the data, authored or reviewed drafts of the paper, provided the strain of Citrobacter rodentium, and approved the final draft.
- Hongyu Zhao, Junyan Qu and Yanan Xia performed the experiments, authored or reviewed drafts of the paper, contributed to the AWR scores implemented, and approved the final draft.
- Yanqing Li conceived and designed the experiments, analyzed the data, authored or reviewed drafts of the paper, and approved the final draft.

### Animal Ethics

The following information was supplied relating to ethical approvals (i.e., approving body and any reference numbers):

Shandong University Laboratory Animal Center and the Ethics Committee on Animal Experiment of Shandong University Qilu Hospital (animal experiment proof certificate number: Dull-2020-11) provided full approval for this research.

## Data Availability

The raw measurements and16S gene pyrosequencing data are available at NCBI: BioProject PRJNA698375. The sample name and group information are available in Table S1.

## Supplemental Information

Supplemental information for this article can be found online at http://dx.doi.org/10.7717/peerj.11585#supplemental-information.

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
