# Peer review of "Clostridium butyricum relieve the visceral hypersensitivity in mice induced by Citrobacter rodentium infection with chronic stress"

_PeerJ, doi:10.7717/peerj.11585_

## Round 0.1 · original submission · Minor Revisions

The reviewers consider the study to be worthy of publication, and only some minor revisions are required. Most importantly, please clarify the statistical tests and bioinformatic pipelines, and please address Reviewer 1's questions regarding the broader implications of the study with respect to the role of C. butyrium in the human microbiome. Furthermore, the English is of suitable quality and does not require further editing, and it is not necessary to avoid using "we" as requested by Reviewer 2.

Reviewer 1 ·

Basic reporting

See my comments below

Experimental design

- Original primary research within Aims and Scope of the journal.
N/A, this is an editorial decision from PeerJ.

See my comments below for the rest

Validity of the findings

See my comments below

Additional comments

The focus of this manuscript is to test the therapeutic property of a probiotic in a model of post-infectious Irritable Bowel Syndrome. Two main features of IBS are addressed in this study 1/hypersensitivity and 2/microbiome changes.
This topic is highly relevant in Gastroenterology. PI-IBS is a seductive hypothesis and quite well studied in both basic and clinical aspects. Unanswered questions in the field relate to understand precisely the short- and long-term effect on the microbiome (not only in terms of composition), and why (and how) such changes in microbiome back to 'normal' has some impact on host physiology after the infectious event (e.g. hypersensitivity, hyperexcitability of enteric neurons, unbalance proteolytic balance at mucosal surface). These outstanding questions in the field of PI-IBS are not particularly well addressed in the present paper.
Interestingly, authors focused on Clostridium butyricum, which is not one of the first probiotic that comes into (my) mind. Changes in microbiota taxonomy are somewhat expected and not very original over what is published in the field. Behavioral score for hypersensitivity is valid, but definitely not state of the art.

Overall and specific comments:
Language: The manuscript reads well, the EN language is good and I found no obvious typos. This is appreciated. I would not request any additional reviewing on that aspect.
Ethics: This study uses animals, specifically mouse model. Ethical aspects seem to be in place, with protocols and SOPs authorized by the host institution.
Data availability: I have been able to access to all 96 replicates from this study in PRJNA698375 project. Labeling of samples is explicit enough if any deeper investigation is required. At this step of reviewing, I have consulted but not reprocessed raw data.
- Antibiotics are not therapies recommended for PI-IBS (except at the stage when there is actual infection). I do not understand the clinical rationale of including such group in this paper.
- It is unclear which type of statistical test is used in each Figure. The small paragraph in the Method section is not enough informative.
- It is unclear the bioinformatic pipeline that has been used to perform 16S analysis. In particular Figure 6 panel B.
- This graphical abstract of in vivo studies is appreciated. But I do not understand this comment from the graphic "C.butyricum/antibiotics gavage continued for 1 weeks"
- Figure 2: I am not convinced there is a significant difference (* P<0.05) between C.rodentium only versus C.rodentium+C. butyricum (orange versus magenta?). To me a 2-way ANOVA would have been more appropriated to statistically analyze this type of dataset. How can the authors explain the small difference in response to 0.1ml injection? This small difference is likely to impact the response to other volume distention.
- I always considered C.rodentium infection to have a huge impact in microbiota composition and overall decrease in richness of OTUs. This is expected because microbial niche is vastly dominated by such enteropathogen (although this is usually temporary). This is not the case here. How the authors can explain such difference? Additional references to support this finding would be appreciated.
- I would recommend author to balance the view of C. butyrium as a strictly beneficial microbes. It has been demonstrated elsewhere that such stain can be considered as a pathogen. This is quite important to consider for clinical relevance of using C. butyrium as a human 'drug'
- C. rodentium is rather a good model for attaching/effacing type of E. coli pathovar such as enteropathogenic E.coli (EPEC) but not so much for EHEC/STEC pathovars O157H7.
- line 98: are you sure 5-6 weeks-old B6 mice weight in average 13-15 g?
- Are the necessary steps have been taken to reduce cage effect at the beginning of the experiment? Such as mixing the bedding between all cages?
- Reduction of OTU: In the text, it could be informative to remind the reader how much OTU have been recovered at the basal timepoint.
- Please remove any non-informative details on taxon ordering such as 'norank_f_' ,'_NK4A136_group' or g_norank_f_norank_o'

·

Basic reporting

English language needs editing.
Intro & background to show context. Literature well referenced & relevant.
Structure conforms to PeerJ standards, discipline norm, or improved for clarity.
Figures are relevant, high quality, well labelled & described.

Experimental design

Original primary research within Scope of the journal. Research question well defined, relevant & meaningful. It is stated how the research fills an identified knowledge gap.
Rigorous investigation performed to a high technical & ethical standard. Methods described with sufficient detail & information to replicate.

Validity of the findings

Conclusions are well stated, linked to original research question & limited to supporting results.

Additional comments

Regarding manuscript entitled “Clostridium butyricum relieve the visceral hypersensitivity in mice induced by Citrobacter rodentium infection with chronic stress”, the authors aimed to examine whether the intervention of C. butyricum or antibiotics could affect the etiology of visceral hypersensitivity.
The study is interesting and well designed. However, minor revision is needed before its acceptance.
Add the aim of the study in the abstract at the end of background subsection.
Add clear hypothesis at the end of introduction before the aim of the study.
Clarify the method used for statistical analysis in each parameter (Student’s t-tests or one-way ANOVA).
Language editing is important before its acceptance.
Please avoid using “we”

---

## Round 0.2 · accepted · Accept

The reviewers are satisfied with the revisions; thank you for addressing their comments.

Reviewer 1 ·

Basic reporting

See my final report below

Experimental design

See my final report below

Validity of the findings

See my final report below

Additional comments

Authors have revised the manuscript well according to my initial comments, with modifications highlighted in the text. As a minor comment, I am not much convinced by the rebuttal for Figure 2 statistics. That being said I will let the scientific community to make their own judgment on this particular data and interpretation.